# Optimising instructional materials for Covid-19 rapid tests for self-sampling and testing: Mapping the optimization process of manufacturer's instructions for use for self-testing RDTs intended for low-literacy contexts

**Moses Kelly Kumwenda** [1,2¤]*, **Madalo Mukoka**[2], **Elena Reipold-Ivanova**[3], **Owen Mhango**[2], **Yasmin Dunkley** [4], **Florence Abok**[3], **Euphemia Sibanda**[5], **Constancia Watadzaushe**[5], **Elizabeth L. Corbett**[4], **Augustine Talumba Choko**[5,6]

1 Malawi Liverpool Wellcome Trust Clinical Research Programme, Blantyre, Malawi, 2 Helse-Nord Tuberculosis Initiative, Kamuzu University of Health Sciences, Blantyre, Malawi, 3 FIND, Geneva, Switzerland, 4 London School of Hygiene & Tropical Medicine, London, United Kingdom, 5 CeSHHAR, Harare, Zimbabwe, 6 Liverpool School of Tropical Medicine, Liverpool, United Kingdom

¤ Current address: School of Primary Care, Population Sciences and Medical Education, University of Southampton, Southampton, United Kingdom

* kumwenda@gmail.com

## Abstract

Simple and easy to use kits for SARS-Cov-2 self-testing during epidemic waves are needed to optimize diagnostic capacity in low- and middle-income countries. SARS-Cov-2 self-testing kits are available, but application of these novel diagnostic technologies is less understood in low and middle-income contexts. We investigated the ability to understand and perform instructions for use (IFUs) for STANDARD Q COVID-19 Ag Test (SD Biosensor) and Panbio COVID-19 Ag Rapid Test Device (Abbott Rapid Diagnostics) for anterior nares (AN) nasal self-sampling and self-testing for COVID-19 in rural and urban Malawi. Qualitative research methods using iterative cognitive interview approach was used to investigate the ability of healthcare providers and lay community members to understand and perform a COVID-19 self-sample or self-test using the manufacturer's instructions for use. A total of 120 iterative cognitive interviews were done with healthcare providers and lay community members for self-sampling (N = 76) and self-testing (N = 44). Cognitive interviews began with the manufacturers version of instructions for use followed by subsequent iterations to refine problematic instructions. Structured interview guide and an observation checklist were used to collect data which was then coded inductively. A framework analysis approach was used to synthesize qualitative data. Study participants were generally proficient at performing a COVID-19 self-sampling and self-testing using the two COVID-19 Rapid Testing Devices. Several of design and content problems within manufacturer's instructions for use made their contextual application sub-optimal. Overall, participants experienced difficulties because of the omission of essential elements within instructions, use of short texts/phrase

**Data Availability Statement:** All 120 qualitative data files and data summaries are available from the Qualitative Data Repository database. https://data.qdr.syr.edu/dataset.xhtml?persistentId=doi:10.5064/F6H1QLNL.

**Funding:** This research was funded by UNITAID (grant number KFW P09022-00) through the Foundation For Innovative New Diagnostics. The funder had no role in the study design, data collection, analysis, decision to publish or preparation of the manuscript.

**Competing interests:** The authors have declared that no competing interests exist.

or lack of a word instruction, the lack of labels on where to open the package; the inconsistencies between word instructions within the instructions for use and the physical contents of the test package; the inability to digest and apply certain technical concepts and the lack of clarity in the phrasing of some text instructions. As expected, healthcare providers experienced fewer problems compared to lay community members. The refinement of these instructions greatly improved comprehension among lay community members. Self-sampling and self-testing for COVID-19 can be performed lay community members with fidelity in a scaled context if the manufacturer's instructions for use have been refined and tailored to the context. In the current study, we have used the study findings to map the optimisation process of manufacturer's IFU'S for self-testing RDT's intended for low literacy contexts including Malawi.

## Introduction

Self-testing products are increasingly becoming widely available and are making enormous contributions within the self-care space to foster universal health coverage while decongesting health infrastructure by shifting screening tasks to lay users. Self-testing products are valuable in recognizing medical conditions early which is important for early mitigation of disease progression and prevention further spread of infection. These technologies are also very useful in context of emerging epidemics such as COVID-19 whose global spread and impacts crippled even the most robust health systems in high income countries [1, 2]. The epidemic imposed an enormous strain on health systems where hospitals run well over capacity, facing shortages of critical care medical resources and personal protective equipment. In the context of this epidemic, the use of specialised test—the Reverse transcription–quantitative polymerase chain reaction (RT–qPCR) tests—failed to cope with widespread disease burden especially in resource poor [3]. Additional layers of access barriers to using such specialized tests included the requirements for skilled laboratory personnel based in centralized facilities, shortages of essential testing supplies, high costs, exorbitant testing user fees, and logistical challenges and poor turnaround times of test results [4, 5]. Clear, simple, and innovative diagnostic technologies that could be rapidly deployed and put to scale were needed to circumvent these underlying access barriers.

Antigen lateral flow rapid tests (Ag-RDT) that directly detect SARS-CoV-2 and with a quick turn-around-time are available [6, 7]. Self-sampling and self-testing using such tests may increase access to timely testing and decrease pressure on health systems during Covid-19 peak periods [8]. Studies suggest that laypersons can perform an Ag-RDT self-test where instructions for use (IFUs) are refined to reduce procedural errors and guarantee accurate and safe specimen collection and testing performance [9]. However, manufacturers of these Ag RDTs are usually naïve of conditions within which their technologies are used and the characteristics of the individuals that use their products. In most part, manufacturers provide standard information within the instructions for use (IFUs) to ensure that individuals safely perform the self-test. To ensure that minimize the possibility of errors and to ensure that users understand how to use the products correctly, it is important to ensure that the design and language of these instructions considers characteristics of the contexts and intended users. Limited data is available on how healthcare providers and lay community members can perform a self-sample or a self-test for SARS-CoV-2 using Ag RDTs in a resource poor and low literacy context. We qualitatively assessed the appropriateness of anterior nares (AN) nasal self-

sampling (where only procedures for collecting the sample are conducted) and self-testing (where participants collect the sample, conduct the test, and interpret the result) using i) STANDARD Q COVID-19 Ag Test (SD Biosensor) and ii) Panbio COVID-19 Ag Rapid Test Device (Abbott Rapid Diagnostics) in Blantyre, Malawi.

## Theoretical framework

Proctors' heuristic taxonomy and conceptualization of implementation outcomes guided the research methods. Proctor et al. describes three distinct theoretical domains in implementation research namely 1) implementation outcomes; 2) service system outcomes and 3) clinical treatment outcomes [10, 11]. The study focused on 'implementation outcomes' of the taxonomy which are defined as "effects of deliberate and purposive actions to implement new treatments, practices, and services" to determine the level of success of the implementation process. Seven categories of implementation outcomes domain include acceptability; adoption; appropriateness; costs; feasibility; fidelity; penetration and sustainability. In this paper, we focused on improving IFUs to ensure that they were appropriate to improve performance fidelity. Our analysis focused on the appropriateness of IFUs for both self-sampling and self-testing performed by healthcare providers and lay community members. Appropriateness was viewed as the 'perceived fit, relevance, or compatibility of the IFUs for COVID-19 self-sampling and self-testing by healthcare providers and lay community members [10].

## Methods

A longitudinal qualitative design was used to iteratively assess the appropriateness of 1) STANDARD Q COVID-19 Ag and 2) Panbio COVID-19 Ag Rapid Tests manufacturer IFUs for self-sampling and self-testing. Cognitive interviews were employed to collect data for optimizing manufacturers IFUs intended for application using self-sampling and self-testing approached. We have previously used cognitive interviews for optimizing IFUs for HIV self-testing [12] and HCV self-testing [13]. The IFU optimization process began with translating the manufacturer IFUs into Chichewa—a local language commonly used in the study area. Then, the IFUs were used on study participants drawn from healthcare providers and lay community members from a rural primary health facility (Lirangwe Health Centre) and an urban tertiary health facility (Queen Elizabeth Central Hospital) in Blantyre. Cognitive interviews were performed in urban and rural Blantyre District, two contexts with differing literacy levels, to promptly determine how manufacturer's IFUs would perform among trained health providers and lay intended users. Healthcare providers extensive experience using rapid diagnostic tests (RDTs) was critical to provide valuable insights on how IFUs could further be improved for use by untrained community members. Study participants from the community provided important information on how lay people understood IFUs and performed a COVID-19 self-sample and self-test.

## Sampling and participant recruitment

To recruit study participants, clearance was sought from the Hospital Director for Queen Elizabeth Central Hospital, the District Health Officer (DHO), and Facility Manager for Lirangwe Health Centre. On commencement of recruitment, clinic contact persons for the study referred potential eligible healthcare providers to a study team member. Purposive sampling approach [14] for both self-sampling and self-testing using STANDARD Q and Panbio COVID-19 Ag Rapid Tests was led by OM who liaised with healthcare providers at health facilities, in outpatient's departments, or community health workers to identify eligible participants. Six trained research assistants (female n = 2; male = 4) provided information about the

study to potential individuals and invited them to participate. Healthcare providers were included if they had prior experience in using RDTs, aged 18 years or older; and lay counsellors and willingness to provide a written informed consent. Individuals from the community were eligible if they were 18 years or older, willing to provide written informed consent; demonstrated functional literacy (i.e. able to read instructions in Chichewa or English) and feeling well enough to comfortably perform study activities.

## Sample size

For each iteration, we planned to recruit at least 4 participants, but the actual number of participants recruited varied depending on attainment of information saturation. Iterations for self-sampling and self-testing for both STANDARD Q and Panbio COVID-19 RDTs ranged between 1 and 6. In total, 120 participants were recruited for self-sampling (n = 76; i.e., healthcare providers = 36; community members = 46) and self-testing (n = 44) components of the study (Table 1). The sample comprised 70 community members and 50 healthcare providers (n = 50) recruited between 16th July 2021 and 18th January 2022. Recruited participants from the community represented a spectrum of intended users for COVID-19 self-sampling and testing.

In the self-sampling component, 31 out of 36 healthcare providers were recruited from Lirangwe health centre with the postponement of recruitment from Queen Elizabeth Central accounting for this difference. There were 47 female and 29 male study participants during self-sampling. In total, an equal number of study participants (n = 38) were recruited for STANDARD Q COVID-19 Ag and Panbio COVID-19 Ag RDTs. Cognitive interviews for both RDTs were done concurrently during all the 6 self-sampling iterations.

In the self-self-testing component, 24 study participants were interviewed for STANDARD Q COVID-19 Ag and 20 participants for Panbio COVID-19 Ag. Cognitive interviews began with STANDARD Q COVID-19 Ag participants before Panbio COVID-19 Ag because the shipment delays of STANDARD Q COVID-19 Ag test-kits. Of the 44 study participants recruited, 28 were community members. An equal number of healthcare providers (n = 8) were recruited from Queen Elizabeth Central Hospital and Lirangwe health centre. There were 23 males and 21 females recruited for the self-testing component. Procedures like those used for self-sampling study were followed during self-testing involving 4 iterations for STANDARD Q COVID-19 Ag and 1 iteration for Panbio COVID-19 Ag. Out of 167 individuals who were screened, a total of 47 declined participation. Non-participation by health providers was because of lack of time due to the increased workloads. Non-participation from

**Table 1. Demographic characteristic of participants.**

| Category | Sub-category | Count | Percent |
|---|---|---|---|
| **Sex** | Male | 52 | 43.3% |
| | Female | 68 | 56.7% |
| **Facility** | QECH | 26 | 21.7% |
| | Lirangwe | 94 | 78.3% |
| **Participant** | Health provider | 50 | 41.7% |
| | Community members | 70 | 58.3% |
| **Test Device** | Standard Q | 62 | 51.7% |
| | PanBio | 58 | 48.3% |
| **Mode of test** | Self-sample | 76 | 63.3% |
| | Self-test | 44 | 36.7% |

community member was due to the fear that a nasal swab can be a source of COVID-19 infection; fear of pain when collecting a nasal sample; that COVID-19 perception that it was not real; a perception that COVID-19 designed for people; and fear that a positive result would imply being secluded from the society.

## Data collection

Starting with the translated version of manufacturer IFUs, a series of iterative cognitive interviews were conducted to refine IFUs for self-sampling and self-testing. A structured cognitive interview guide that reflected the steps depicted in the manufacturers IFUs was used. Data collection began with healthcare workers before members of the community. Six experienced and well-trained qualitative researchers (Male n = 4; Female n = 2) led by OM (BSoc) and MK (PhD) conducted cognitive interviews following the structure of the IFUs. All the six data collectors had post-high school and graduate qualifications and had accumulated more than 6 years average qualitative research experience including cognitive interviewing. Study participants did not know the researchers prior to the study as the relationship with researchers was established during the study. Researcher's biases and assumptions were somehow influenced by prior research in the area HIV self-testing, but this experience was important in shaping the direction of this study. Researchers went step-by-step asking participants how they understood each instruction and then executing the instruction under observation. Interviewers requested study participants to (1) read the text and observe the pictorial instruction, (2) reflect and explain how they understood the instruction, (3) perform the actions the instruction required, and (4) reflect on possible changes to improve comprehension. Scripted probes embedded in the structured guide questions and spontaneous probes enhanced the depth of collected data. Daily debriefings that were designed to gain immediate insights from the field and promptly use them to revise the IFUs increased the thoroughness of data collection process.

All cognitive interviews were audio-recorded using digital audio recorders. Study participant responses and observed performance of instructions were also methodically documented using an observation checklist. Research assistants drafted detailed field notes after each cognitive interview to allow nuanced understanding of data collected using the observation checklist. Within each group (healthcare providers, community members, rural, urban), iterative cognitive interviews continued until no new suggestions (saturation of information) for improvement emerged [15]. Changes made to IFUs during each iteration were communicated to FIND through weekly interactive meetings. Samples collected during the self-sampling substudy were not tested for COVID-19 but discarded according to the study Specimen Handling and Biosafety procedures.

## Data analysis

Recorded interviews were transcribed and translated into English and saved on password-secured computers at the Helse Nord TB Initiative (HNTI) of the Kamuzu University of Health Sciences (KUHES). Data analysis was performed during the incorporation of the changes into the IFUs and consolidation of findings at the end of data collection. Findings reported in this paper are those made from the analysis done after all data was collected. After each interview, fieldnotes were transcribed with attention to instructions that participants found ambiguous. Through daily debriefings, researchers mutually reviewed observation notes, audio recordings and checklists to finalise a list of suggested changes. Study team members debated findings on suggested changes before incorporating these into new version of the IFU. When no new suggestion emerged within a round of interviews, a full list of recommended improvements was compiled, and the study team agreed on specific improvements to

be incorporated within the current version of IFUs. Data was coded inductively using a Framework analysis approach consistent with the steps outlined in the testing process for each test. Proctor framework of implementation outcomes informed the study methodology and the analytical strategy with emphasis on the appropriateness dimension [10].

## Ethical considerations

Ethical clearance and approvals were sought from the College of Medicine Research Ethics Committee (COMREC) [Ref P.03/21/3277]. All study participants provided either a written or thumbprint informed consent.

## Findings

A variety of perspectives were expressed on design and content problems within the manufacturer's version of the IFUs made self-sampling and testing using STANDARD Q COVID-19 Ag and Panbio COVID-19 Ag Rapid Tests problematic especially among untrained community members. Identified design and content problems aided the process of optimizing instructional materials to enhance their appropriateness within the implementation context. Five broad themes emerged from the analysis on the key problems undermining the ability of lay community members and healthcare providers to self-sample and self-test for COVID-19 using the two test-kits. These themes were 1) participant inability to identify some kits content; 2) omission of some essential elements; 3) incoherence between description within instructions and actual contents of a test package; 4) difficulties to understanding and applying of technical concepts and 5) diminished clarity in the wording of text instruction.

### Inability to identify kit contents

A common view amongst interviewees was that the organization of IFUs and the packaging of the test contents for both STANDARD Q and Panbio COVID-19 Ag Rapid Tests were sometimes not synchronized with each other, and this created problems on comprehension and interpretation of the instructions. During the self-sampling study, contents of both tests were provided after several instructions, and this disrupted the logical sequence of instructions. Data demonstrated that participants required descriptions of kits contents/components earlier before any instruction to enable them simply to identify them. Introducing kits components later within IFUs made study participants struggle to recognize certain components when mentioned for the first time. Graphic illustrations within manufacturers IFUs also needed to match the actual component of the test kit.

*Q1 'There is a need for the instruction especially the section showing kit contents to show the solution tube holder as well as the plastic film that are not indicated on the original IFU.'*

(P01, STDQ, Self-sample, QECH)

*Q2. 'Some items found in the bag are not indicated on the kit contents i.e. the film and the tray holder.'*

(P04, STDQ, Self-Sample, Lirangwe)

For example, on kits contents within both tests, intended users saw an illustration of unwrapped test device. However, when they opened the test package, they uncovered a foil package (See Q1 and 2). Such incongruities made manufacturers versions of IFUs ambiguous and confusing to study participants. For the STANDARD Q COVID-19 Ag test, some study

participants were unable to recognize the actual 'well' on the test device where they could apply the specimen. Through the iterative processes of Cognitive Interviewing, instructions for both tests were reordered to fit the purpose and context. Pictures were improved to guarantee that both the foil package and the test device were shown within pictorial representations for the Panbio COVID-19 Ag Rapid Test. Specific to a problem of the inability for users to recognise a 'well' on a test device of STANDARD Q, the word 'round' was introduced and added to the instruction 8 to enhance comprehension.

## Omissions of essential elements

In the manufacturers IFUs for both tests, study participants observed several omissions that made instructions challenging to grasp. A recurrent theme in the interviews was a sense amongst interviewees that certain essential elements that were important for enhancing comprehension were excluded or missing. Several omissions were identified within both pictorial and text instructions for both test kits. Within both rapid tests text instructions, frequent use of short phrases was insufficient to convey the full meaning of the instruction. In self-sampling and self-testing instructions, study participants underlined the importance of including handwashing with soap within the first/second instruction on sanitisation of hands. The opening of some of the packages was equally difficult to perform due to the lack of labels for notifying users on how and where to open the pouches.

> Q3. *'The instruction should also add a statement instructing people to sanitize their hands and not wash hands alone.'*
>
> (P11, PanBio, Self-Sample, Iteration 2, Lirangwe)
>
> Q4 *'Add the use of hand sanitizer when washing hands not water only.'*
>
> (P14, STDQ, Self-Sample, Iteration 2, Lirangwe)

Within the Panbio COVID-19 Ag Rapid Test instructions for self-sampling, some instructions were not numbered to provide a sequential step-by-step flow of the instructions. For example, having no instruction on washing hands. Omission of important instruction disrupted the chronological flow of the instructions making it challenging for some participants to recognize the order of instructions. Further, Panbio COVID-19 Ag test instruction on checking the expiry date of the test device did not have a corresponding picture of where users find the expiry date on the test package (see Q5). For STANDARD Q COVID-19 Ag, instruction 5 (on *opening the bottle*) omitted including a detail that the bottle contained buffer solution. Inclusion of the word buffer in the instruction minimised the likelihood of inadvertent spillage of the buffer solution when opening the bottle. Again, for **instructions 8** (on inserting *the swab into a solution tube*) and **10** (on *applying 4 drops of extracted sample to the sample well of the test device*) for STANDARD Q COVID-19 Ag self-testing, we added word instructions guiding intended users to '*place the swab aside after removing it from the solution tube*' and place the '*solution tube aside after applying the drops onto the test device*' respectively. The interpretation of results section for STANDARD Q COVID-19 Ag test during self-testing only had pictorial instructions without word instructions probably because the instructions were designed for professional use. But still, even health workers indicated the need to have text instructions which we added to augment user comprehension.

> Q5. *'There is a need for the instruction to include a picture of the test device especially where expiry dates are indicated to help people identify the expiry dates easily.'*

(P01, STDQ, Self-sample, Iteration 1, QECH)

Q6 'Add a statement emphasizing "pa kabotolo la timadzi"[a bottle containing solution] in the Chichewa IFU'

(P22, STDQ, Self-Sample, Iteration 3, Lirangwe)

## Inconsistencies between IFUs and test package contents

Contradictions between the manufacturer's IFUs and the actual contents of materials packaged together with the test devices was a recurrent theme in the data. The STANDARD Q COVID-19 Ag test for example, **instructions 2, 5** and **6** pictorial illustrations contained pictures of hands in gloves for both self-sampling and self-testing studies. In their accounts of the events surrounding the lack of gloves in the packaged materials, interviewees questioned how they could perform a test without using gloves. Also, the manufacturers only presented pictorial representations of the test device, or buffer tube without showing their protective pouches within an instruction on the content of the kits (see Q6 and Q7). This inconsistency spawned a great deal of confusion as participants physically saw the protective package of the test device and a buffer tube. We therefore incorporated pictures of the protective pouches to improve comprehension of this instruction. For both self-sampling and self-testing for Panbio COVID-19 Ag test, we also supplemented pictures showing the foil pouch of the test device to make certain that participants understood where to locate the test device.

Q7 'The pictures of the test device shown be shown on the IFU whilst in its protective plastic bag so that it should be clear and easy for one to identify it easily.'

(P14, PanBio, Self-Sample, Iteration 2, Lirangwe)

Q8 'The instruction should be rephrased so that it should instruct people to open the protective bag for swab stick and remove the swab stick from the bag . . .'

(P06, STDQ, Self-Sample, Iteration 1, Lirangwe)

During self-testing, Panbio COVID-19 Ag test word **instruction 2** was altered from inviting users to 'carefully open the kit box as it will be used later on' which was included in a professional use kit used during the self-sampling study. This instruction was different from the manufacturer's version of the IFUs found inside the self-test kit box during the self-testing study. The wording was improved to reflect the self-testing packaging of the test kit content and read: 'open the kit box and remove each of the components to perform a single test'. Additionally, a picture of a tube holder attached to the inside of the kit box as shown on the manufacturer's version of the IFUs received before the arrival of Panbio COVID-19 Ag self-test kits was replaced with a picture of a new tube holder.

Several instructions required additional details to be clearer to interviews as short phrases or lack of sufficient pictorial illustrations rendered such instructions fail to amply convey the envisioned meaning. In such scenarios, additional text was included, and existing pictures modified (i.e. including new pictorial instructions or labels) to improve clarity and ensure that the instruction relayed the required meaning. STANDARD Q COVID-19 Ag test for example, some participants found the process of applying 4 drops onto the test device challenging in **instruction 8** because of the lack of the word 'squeeze' in this instruction (see Q8). The word 'squeeze' was added in the second picture of instruction 8 to emphasize the importance of squeezing the swab through the walls of solution tube to ensure that enough of the sample was

retained in the tube. This word was also added to the pictorial **instruction 10** on the application of the sample onto the test device.

> *Q9 'The instruction has to clarify that one need to squeeze the bottle with enough force to extract liquid from the swab stick.'*
>
> (P01, STDQ, Self-Sample, Iteration 1, Lirangwe)

There were some circumstances which necessitated splitting one instruction into two parts to improve intelligibility while in other instances, some words were subtracted from instructions to enhance consistency with the pictorial instructions. STANDARD Q COVID-19 Ag test **instruction 8** for example, the word instruction denoted that the user should *'swirl the solution inside the tube for more than 10 times'* while the pictorial instruction only showed *'10x times' (see Q9 and Q10)*. To ensure coherence between the two, we amended the word instruction to *'stir the solution inside the tube 10 times'*. For Panbio COVID-19 Ag test, the manufacturer version for self-testing of **instruction 3** stated that the buffer bottle should be opened by *'twisting and pulling the tab'* which was incompatible with practicalities on the ground (see Q11). Most study participants observed that by only *'twisting the tab'*, the instruction was sufficient since *'pulling'* was not necessary as the tab separated from the buffer bottle immediately after *'twisting'*.

> *Q10 'The instruction should have a written statement instructing people to swirl 10 times.'*
>
> (P05, STDQ, Self-Sample, Iteration 1, Lirangwe)
>
> *Q11 'The instruction should clarify that people should insert the swab stick in the bottle and hold the bottle with one hand and use the other hand for swirling.'*
>
> (P09, STDQ, Self-Sample, Iteration 1, Lirangwe)
>
> *Q12 'The instruction has to be rephrased for it to make complete sense because as it is only medical personnel can understand it well and not the ordinary person.'*
>
> (P02, PanBio, Self-Sample, Iteration 1, QECH)

Similarly, Panbio COVID-19 Ag test **instruction 4** on pouring liquid from buffer bottle onto the tube, the manufacturer versions specified adding at least two drops to the fill-line of the tube (see Researcher Observation 1). In practice and during self-sampling, more than two drops were in most cases needed to get to the fill-line of the tube. However, the buffer bottle for self-testing contained buffer solution that was sufficient to reach the fill-line or slightly above it when all of it was poured into the tube. Complicating this was that two participants did not recognize where the fill-line was (See Researcher Observation 2). Thus, the instruction to *'add at least two drops'* did not add any value to both the self-sampling and self-testing processes. The instruction was reworded to correspond with users' experiences during the self-sampling and self-testing stages. In terms of sample collections for the Panbio COVID-19 Ag test, there were confusions introduced by the wording of **instructions 7 and 8** on swabbing the nostrils because much of the information on specimen collection process was contained in **instruction 7** while **instruction 8** simply repeated what was described in **instruction 7**. To optimise comprehension, **instruction 8** was removed and text was added to instruction 7 to capture the content of instruction 8 (See Researcher observation 3).

**Researcher Observation 1**: The participant squeezed the buffer bottle more than two times as per instruction because it was impossible for the buffer to reach the fill line by squeezing the buffer. (P10, PanBio, Iteration 2, Lirangwe)

**Researcher Observation 2**: The participant had challenges in identifying the fill line on the tube with buffer. (P13, PanBio, Iteration 3, Lirangwe)

**Researcher Observation 3**: The participant suggested that . . . we should remove the note below instruction 7 since it is confusing with what instruction 8 is stating. (P07, PanBio, Iteration 2, Lirangwe)

## Understand and applying technical concepts

A variety of perspectives were expressed recharging the difficulties experienced in understanding certain technical concepts contained in both self-sampling and self-testing IFUs for both tests. These concepts needed to be toned down or even contextualised to guarantee that untrained users grasped the underlying meaning when performing self-sampling and self-testing. For example, it was difficult for study participants to understand *'inserting the swab into nostrils for 2cm'* (STANDARD Q COVID-19 Ag test) and *'1.5cm'* (Panbio COVID-19 Ag) to collect a nasal sample. This was so because of the lack of a measuring device coupled with the inability of some individuals to concretise these abstract measurements (See Q12 and Q13). Early during the self-sampling sub-study, the inability to understand these parameters made some study participants experience problems when collecting a nasal sample. To enhance comprehension, the study improvised by using *'the length of a thumbnail'* as a proxy for the length required to insert the swab in the nostril for both tests.

Q13: 'The swab stick should have a line to make 2cm so that people should know the appropriate length of inserting the swab stick in the nostril'

(P01, PanBio, Self-Sample, Iteration 1, QECH)

Q14: '1.5cm should be shown on the swab stick.'

(P01, STDQ, Self-Sample, Iteration 2, Lirangwe)

For the STANDARD Q COVID-19 Ag test, the manufacturers version of IFUs used during self-sampling did not contain sufficient wording about checking the expiry of the test device using the desiccant colours crystal in **Instruction 3**. In Malawi, it was difficult to translate the word *'crystals'* to ensure that the original meaning of this word was retained. The study team simplified this instruction by the word *'sand'* to represent crystals. The instruction was further broken into smaller parts to enhance comprehension and include information on what to do when the device is expired (See Q14). However, colour coding of the crystals used by the manufacturer was not consistent with the universally recognised traffic light colour codes. The STANDARD Q COVID-19 Ag test manufacturer used 'yellow' colour to represent 'not expired' and 'green' to represent 'expired'. Participants suggested the use of yellow to represent 'expired' and green to represent 'not expired' but implementation of these was difficult for researchers (See Q15). The study recommended the manufacturer to modify the colour coding scheme to correspond with the universally accepted norm. Additional text was included to ensure that intended users found it easier to grasp this unorthodox colour code.

Q15: 'The instruction should indicate that inside the bag containing the test device there is also another small bag showing green or yellow sand.'

(P07, STQ, Iteration 2, Lirangwe)

Q16: 'The instruction should clarify what people should do if they find that the test device has expired i.e. whether to proceed with the testing or not'

(P01, STQ, Iteration 1, QECH)

## Diminished clarity in the wording of text instruction

A recurrent theme in the interviews was a sense amongst interviewees that the phrasing of several instructions made some study participants struggle to discern the meaning or perform the intended task that the instruction required. In some cases, text used, or the composition of the words did not correspond with simple actions needed to successfully execute the instruction. Several instructions for both tests required participants to open several pouches such as a foil pouch containing a test device and a swab but without describing how. The research team included a statement describing exactly what users should do to open the pouches such as '*open by tearing the pouch or open where there is an arrow by peeling the plastic paper*'. Again, **instruction 2** of STANDARD Q COVID-19 Ag test required users to '*open the test device pouch and __check__ for the test device and desiccant pack.*' Participants suggested that the instruction needed to specify that users should '*open the test device pouch and __remove__ the test device and desiccant pack*' from the pouch and not just checking. Clarity was enhanced by adding text or proving the pictorial instructions.

Some instructions did not state exactly where users should place certain components of the test. For example, manufacturers version of **instruction 5** for STANDARD Q COVID-19 Ag test for self-testing worded: '*fix the solution tube*' was reworded based on participant suggestions to '*fix the tube _here_*' to enrich comprehension of where to place the tube after opening the tube. **Instruction 7** of the same test kit was improved from '*insert the sterile swab and rotate for both nostrils*' to '*insert the sterile swab and rotate 10 times in each nostril*' to emphasise the number of times one was required to rotate the swab which was only present in the pictorial instruction (see Q16). Additionally, applying enough buffer solution to a bottle or buffer solution containing a sample to a test device was practically tricky to some study participants because of the wording of the instruction.

Q17 'The instruction should emphasize that one should swirl the swab stick in each nostril 10 times and the instruction should clarify that the swab stick must be swirled both nostrils.'

(P01, STQ, Iteration 1, QECH)

The phrasing of some instructions made the translation of such word or text fail to capture a true meaning of the instruction. For example, manufacturer version of **instruction 11** of Panbio COVID-19 Ag stated that '*secure the tube with the blue cap*'. English and translated version of the instruction was misleading to participants. We revised the instruction to '*close the tube with the blue cap*' to enhance comprehension.

Q18: 'The participant suggested that the instruction should instruct people to close the blue cap tightly to avoid development of the bubbles in the bottle that can lead to invalid.'

(P06, PanBio, Self-Sample, Iteration 1, Lirangwe)

In the self-testing study, most participants did not experience any difficulties in understanding how test results should be interpreted for both tests. They also interpreted their self-

test results accurately. However, there were few participants who had problems to understand an invalid result (PanBio N = 1), a positive result (PanBio N = 1), and a Negative result (PanBio N = 1)

## Discussion

To our knowledge, this study is first to describe a concurrent optimization process of IFUs for STANDARD Q COVID-19 Ag Test (SD Biosensor) and Panbio COVID-19 Ag Rapid Test Device (Abbott Rapid Diagnostics) in Malawi to ensure that they are appropriate for the context. It is important to first highlight that assessment of acceptability and safety these Covid-19 RDTs have been favorable [16, 17]. Consistent with WHO recommendation for usability studies of RDTs, this study assessed the ability of intended users to correctly comprehend key messages from packaging and labelling [18]. This study found that manufacturers IFUs less compatible to the contexts since community members experienced challenges with the kit content, omission of essential elements, disagreements between instructions and contents of a test package, understanding technical concepts and clarity of instructions wording. The use of translated manufacturers IFUs (pictorial and written materials) could not sufficiently support self-sampling and self-testing of self-collected anterior nares samples in Malawi which were also observed during HIV self-testing [12]. A study in rural Sudan reported end-users experiencing some difficulties to interpret IFU's for two Malaria RDTs and this affected performance [19]. Clearly, the application of the RDT for self-testing or self-sampling require in-depth understanding of the interaction between IFUs and the context and an added layer of dealing with contextual issues that affect comprehension and subsequently performance.

Findings presented here demonstrate the appropriateness of anterior nares nasal self-sampling and self-testing following the optimization process as observed in a follow-up study which reported high feasibility and acceptability [20]. The optimization process laid the ground for possible scaled application for COVID-19 self-testing. As projected, we established that translated IFUs for STANDARD Q COVID-19 Ag Test (SD Biosensor) and (ii) Panbio COVID-19 Ag Rapid Test were understood quite easily among health care providers when compared with lay community members for both self-sampling and self-testing as also reflected by our feasibility evaluation [20]. HIV self-testing studies have previously exhibited that compliance to instructional materials and performance in terms of execution of self-testing and interpretation of results is frequently elusive in populations with low literacy [21–23]. During the self-sampling study, participants had more performance problems with STANDARD Q COVID-19 Ag Test than Panbio COVID-19 Ag Test because its instructions were more streamlined. Interviewees were having fewer performance issues with both tests during the self-testing because improvements to contextual issues proposed during the self-sampling phase had considerably refined the instructions designed for self-testing. The preliminary observations on appropriateness obtained from the self-sampling performance were conveyed to manufacturers through FIND and contributed to further improvements of the IFUs during the self-testing phase.

Refining IFUs for COVID-19 self-sampling and self-testing to optimise appropriateness unearthed several textual and pictorial instructions that were challenging to comprehend and contributed towards poor compliance to IFU procedures, an important element for guaranteeing fidelity, accuracy, and safety. Consistent with these findings, Lindner et al (2021) also observed deviations from COVID-19 self-testing occurring during sample collection, sample extraction, and sample application on the test device [9]. To address these problems, we used suggestions from interviewees to inform the IFUs refinement process to generate an optimal version. For scaled implementation of COVID-19 self-sampling and self-testing or self-testing

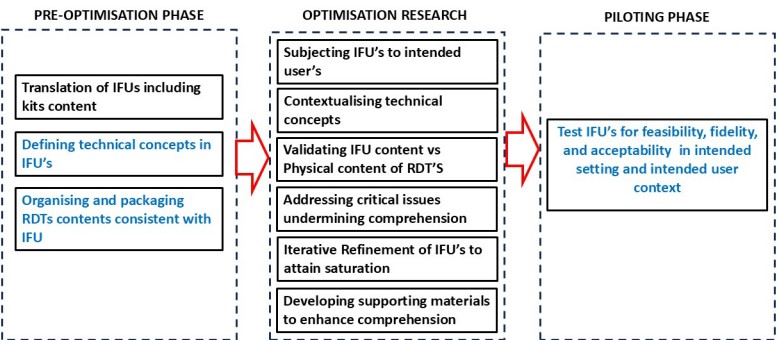

**Fig 1. Optimisation process of manufacturer's IFU'S for self-testing RDT's intended for low literacy contexts.**

for similar epidemics, we suggest the use of additional information and support during initial implementation periods to optimize lay user confidence especially those within the lowest literacy bracket. A recent qualitative study has demonstrated that the provision of training to intended users improve their confidence to perform a covind-19 self-test [24]. Training lay people, on how to perform a self-sampling or self-testing for COVID-19 may be another option of optimise IFU appropriateness and fidelity. As correctly observed by Gupta-Wright [25], knowledge and awareness of self-testing obtained through HIV self-testing simplified the challenges that lay users experience with interpreting test results in this study.

In this study, we suggest a three phased process of optimising IFU's for self-testing RDT's intended for low literacy contexts (Fig 1). In the pre-optimisation phase, it is critical to translate the IFU's to local language; define the meaning of technical concepts contained in the IFU; and ensuring that the content of the items included in the package is consistent with what has been described in the IFU. During the optimisation research phase, it is important to subject the IFU to intended users and intended distributors for these devices. This process should be iterative in nature to attain saturation of information and used to contextualise technical concepts, validating the device contents and order of performing the testing process, addressing important gaps and issues undermining compression and where required, developing supporting materials. Once these steps have been done, then the RDT can be implemented within a pilot context to test acceptability, feasibility, and fidelity among other implementation outcome parameters.

The current study had several limitations namely the different sample sizes for both lay community members and healthcare providers in rural and urban health facilities. Having more lay community members was important because their experiences and views depict what to expect during implementation if COVID-19 self-sampling and self-testing were to be scaled-up. The differences in the sample sizes did little to influence data analysis, since our focus was on how individuals understood each instruction and saturation of information was an important yardstick to determine when to terminate data collection. For the self-sampling phase, the IFUs and kit packaging were done for professional use purposes and were repackaged by the study team. Translations of the manufacturers' IFUs for both test-kits were done by the researchers and this may create an impression that manufacturers IFUs had both versions. Lastly, translated versions of the IFUs were not packaged with the kit. Participants were given these once they had opened the pouch. However, all these limitations did not have a bearing on how health providers and lay community users performed self-sampling and self-testing.

## Conclusions

Health providers and lay users from the community were able to perform COVID-19 self-sampling and self-testing using STANDARD Q COVID-19 Ag and Panbio COVID-19 Ag Test following the IFU optimization process. Iterative adapted cognitive interviews allowed the identification and improvement of several errors that undermined the appropriateness of IFUs and performance fidelity when the test was in the hands of users, especially lay community members. Findings from the study underline specific issues implementers should be aware of to overcome performance and fidelity problems when introducing similar innovation in contexts punctuated with low literacy. In the current study, we have used the study findings to map the optimisation process of manufacturer's IFU'S for self-testing RDT's intended for low literacy contexts. With these findings, recommend the possibility of having "optimized instructional materials" not only for COVID-19 rapid tests but for all commercially available self-sampling and testing products. As described in our previous work, the optimization of performance needs to transcend improvements made to IFUs to encompass physical demonstrations and visual aids in these settings.

## Acknowledgments

The authors would like to thank all the participants and the research team members in Malawi. The Ministry of Health, Blantyre District Health Office, and the management of Queen Elizabeth Central Hospital for providing an enabling environment for data collection. The Helse Nord TB initiative team and pathology department of the Kamuzu University of Health Sciences for providing administrative support to this study.

## Author Contributions

**Conceptualization:** Moses Kelly Kumwenda, Madalo Mukoka, Elena Reipold-Ivanova, Florence Abok, Euphemia Sibanda, Constancia Watadzaushe, Elizabeth L. Corbett, Augustine Talumba Choko.

**Data curation:** Moses Kelly Kumwenda, Madalo Mukoka, Owen Mhango, Augustine Talumba Choko.

**Formal analysis:** Moses Kelly Kumwenda, Madalo Mukoka, Owen Mhango, Yasmin Dunkley, Elizabeth L. Corbett, Augustine Talumba Choko.

**Funding acquisition:** Elena Reipold-Ivanova, Florence Abok, Euphemia Sibanda, Elizabeth L. Corbett, Augustine Talumba Choko.

**Investigation:** Moses Kelly Kumwenda, Madalo Mukoka, Yasmin Dunkley, Florence Abok, Euphemia Sibanda, Constancia Watadzaushe, Augustine Talumba Choko.

**Methodology:** Moses Kelly Kumwenda, Madalo Mukoka, Owen Mhango, Florence Abok, Euphemia Sibanda, Constancia Watadzaushe, Elizabeth L. Corbett, Augustine Talumba Choko.

**Project administration:** Moses Kelly Kumwenda, Madalo Mukoka, Elena Reipold-Ivanova, Yasmin Dunkley, Florence Abok, Constancia Watadzaushe, Elizabeth L. Corbett.

**Resources:** Moses Kelly Kumwenda, Madalo Mukoka, Owen Mhango, Yasmin Dunkley, Euphemia Sibanda, Elizabeth L. Corbett, Augustine Talumba Choko.

**Supervision:** Moses Kelly Kumwenda, Elena Reipold-Ivanova, Elizabeth L. Corbett.

**Validation:** Moses Kelly Kumwenda, Madalo Mukoka, Elena Reipold-Ivanova, Yasmin Dunkley, Euphemia Sibanda, Constancia Watadzaushe, Elizabeth L. Corbett, Augustine Talumba Choko.

**Visualization:** Moses Kelly Kumwenda.

**Writing – original draft:** Moses Kelly Kumwenda, Owen Mhango.

**Writing – review & editing:** Moses Kelly Kumwenda, Madalo Mukoka, Elena Reipold-Ivanova, Owen Mhango, Yasmin Dunkley, Florence Abok, Euphemia Sibanda, Constancia Watadzaushe, Elizabeth L. Corbett, Augustine Talumba Choko.

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
