## [Decision Letter · Decision Letter 0]

12 Mar 2024

PONE-D-24-01686TITLE: Optimising instructional materials for Covid-19 Rapid Tests for Self-Sampling and Testing: Mapping the optimisation process of manufacturers Instructions for Use for self-testing RDT's intended for low-literacy contextsPLOS ONE

Dear Dr. Kumwenda,

Thank you for submitting your manuscript to PLOS ONE. I thank you for your interest in this field of work and following up with a great report. After careful consideration, we feel that it has merit but does not fully meet PLOS ONE’s publication criteria as it currently stands. Therefore, we invite you to submit a revised version of the manuscript that addresses the points raised during the review process. There are several comments from the reviewers which I agree and recommend to be responded appropriately. Correction of the English writing for grammar and typographical errors are therefore needed. Overall I think your report is interesting, novel and very useful. I hope you could refine it as suggested and return at due time.    

We look forward to receiving your revised manuscript.

Kind regards,

Bachti Alisjahbana, MD, PhD

Academic Editor

PLOS ONE

“The authors would like to thank all the participants and the research team members in Malawi. The Ministry of Health, Blantyre District Health Office, and the management of Queen Elizabeth Central Hospital for providing an enabling environment for data collection. The Helse Nord TB initiative team and pathology department of the Kamuzu University of Health Sciences for providing administrative support to this study.  This work was funded by FIND.”

“This research was funded by UNITAID

(grant number KFW P09022-00) through the Foundation For Innovative New Diagnostics. The funder had no role in the study design, data collection, analysis, decision to publish or preparation of the manuscript.”

4. Please include your tables as part of your main manuscript and remove the individual files. Please note that supplementary tables (should remain/ be uploaded) as separate "supporting information" files.

Reviewers' comments:

Reviewer's Responses to Questions

**Comments to the Author**

1. Is the manuscript technically sound, and do the data support the conclusions?

Reviewer #1: Yes

Reviewer #2: Yes

2. Has the statistical analysis been performed appropriately and rigorously? 

Reviewer #1: N/A

Reviewer #2: N/A

3. Have the authors made all data underlying the findings in their manuscript fully available?

Reviewer #1: No

Reviewer #2: Yes

4. Is the manuscript presented in an intelligible fashion and written in standard English?

Reviewer #1: No

Reviewer #2: Yes

5. Review Comments to the Author

Reviewer #1: General overview

This manuscript provides a relevant insight into a crucial aspect of incorporating self-administered test kits for infectious diseases. The findings should provide a reliable guide for manufacturers in developing instructions for use (IFU) that can be easily understood and applied by its intended users. I suggest the following revisions to improve the quality of this manuscript.

2. Section-by-section review

2.1. Introduction

“Clearly, simple, and innovative diagnostic technologies that could rapidly deployed and put to scale were needed to circumvent these underlining barriers.” (lines 83-85). This sentence needs grammatical editing, e.g. clear instead of clearly and adding be between rapidly and deployed, or rephrase the entire sentence to improve clarity.

2.2. Methods

To comply with COREQ, more information about the characteristics and reflexivity of the research team is needed. Additional information about whether there is non-participation (and why) should also be provided.

2.3. Results

"Four broad themes emerged from the analysis...” but the sentence listed five themes (lines 229-235). Please revise accordingly.

Moreover, theme #3 (incoherence between description within instructions and actual contents of a test package) and theme #5 (diminished clarity in the wording of text instruction) conceptually overlaps so some redefining is needed. The choice to use the word “Contradictions” in the heading of section 3 of results (line 309) felt imprecise as the findings and quotes described in that section presented inconsistencies or lack of clarity in the instructions rather than contradictions. The term "incoherence" used previously is more precise.

Reviewer #2: Although Moses Kelly Kumwenda and coworkers' manuscript is in a field between scientific and social fields, it is of great interest, considering that rapid tests in low-literacy contexts could save lives.

Therefore, in the discussion section, the low number of participants is no problem. In my opinion, this is a driven study for future application on other rapid tests. In fact, my advice is to include the possibility of having an "optimizing instructional material" not only for COVID-19 rapid tests but for all commercially available self-sampling and testing tests.

6. PLOS authors have the option to publish the peer review history of their article (what does this mean?). If published, this will include your full peer review and any attached files.

Reviewer #1: No

Reviewer #2: **Yes: **Erika Cione

---

## [Author Response · Author response to Decision Letter 0]

11 Sep 2024

24th April 2024

PLOSE ONE Editorial Office

PLOSE ONE Journal

Dear Editors,

RE: PONE-D-24-01686 R1 - Optimising instructional materials for Covid-19 Rapid Tests for Self-Sampling and Testing: Mapping the optimisation process of manufacturer’s Instructions for Use for self-testing RDT's intended for low-literacy contexts

Thank you very much for pointing out issues in our manuscript that require changing in order to improve it. We have addressed all the issues raised and have provided our point-by-point responses in this letter. We begin by addressing comments raised by the Academic editor then the reviewers.

Academic Editor Comments

Review comment 1: Please ensure that your manuscript meets PLOS ONE's style requirements, including those for file naming. The PLOS ONE style templates can be found at

Response: Thank you very much for your comment. We have structured the manuscript in accordance with PLOSOne formatting style provided by the templates provided.

Review comment 2: Thank you for stating the following in the Acknowledgments Section of your manuscript:

“The authors would like to thank all the participants and the research team members in Malawi. The Ministry of Health, Blantyre District Health Office, and the management of Queen Elizabeth Central Hospital for providing an enabling environment for data collection. The Helse Nord TB initiative team and pathology department of the Kamuzu University of Health Sciences for providing administrative support to this study. This work was funded by FIND.”

“This research was funded by UNITAID

(grant number KFW P09022-00) through the Foundation For Innovative New Diagnostics. The funder had no role in the study design, data collection, analysis, decision to publish or preparation of the manuscript.”

Response: We have removed funding information in the acknowledgement section of the manuscript. We will not update our funding statement. The funding statement will remain the same: “This research was funded by UNITAID (grant number KFW P09022-00) through the Foundation For Innovative New Diagnostics. The funder had no role in the study design, data collection, analysis, decision to publish or preparation of the manuscript.”

Review comment 3: Please include a separate caption for each figure in your manuscript.

Response: We have included separate caption for Fig. 1 in the manuscript. (Page 25; Lines 568-569)

Review comment 4: Please include your tables as part of your main manuscript and remove the individual files. Please note that supplementary tables (should remain/ be uploaded) as separate "supporting information" files.

Response: Thank you very much for your comment. We have included Table 1 in the main manuscript. (Page 8; Lines 166-167)

Review comment 5: Please review your reference list to ensure that it is complete and correct. If you have cited papers that have been retracted, please include the rationale for doing so in the manuscript text or remove these references and replace them with relevant current references. Any changes to the reference list should be mentioned in the rebuttal letter that accompanies your revised manuscript. If you need to cite a retracted article, indicate the article’s retracted status in the References list and also include a citation and full reference for the retraction notice.

Response: Refence list has been reviewed and we are satisfied that it is now complete. No references that we have used has been retracted papers. We checked all the all the reference used on retraction data base (http://retractiondatabase.org/), none has been retracted.

Reviewer #1 Comments

Reviewer #1: General overview

Review comment 1: This manuscript provides a relevant insight into a crucial aspect of incorporating self-administered test kits for infectious diseases. The findings should provide a reliable guide for manufacturers in developing instructions for use (IFU) that can be easily understood and applied by its intended users. I suggest the following revisions to improve the quality of this manuscript.

Response: Thank you very much for the wonderful observation and positive feedback about our manuscript. We have included Table 1 in the main manuscript. (Page 8; Lines 166-167).

2. Section-by-section review

2.1. Introduction

Review comment 2: “Clearly, simple, and innovative diagnostic technologies that could rapidly deployed and put to scale were needed to circumvent these underlining barriers.” (lines 83-85). This sentence needs grammatical editing, e.g. clear instead of clearly and adding be between rapidly and deployed or rephrase the entire sentence to improve clarity.

Response: Thank you very much for your observation. We have changed the sentence according to your suggestion. The sentence now reads ‘Clear, simple, and innovative diagnostic technologies that could be rapidly deployed and put to scale were needed to circumvent underlying access barriers. (Page 4; Lines 86-88)

2.2. Methods

Review comment 3: To comply with COREQ, more information about the characteristics and reflexivity of the research team is needed. Additional information about whether there is non-participation (and why) should also be provided.

Response: we have provided information on the characteristics and reflexivity of the research team in the data collection section (Page 9; Line 193-198). We have also provided information on non-participation in the sample size section (Page 9; 183-187)

2.3. Results

Review comment 4: "Four broad themes emerged from the analysis...” but the sentence listed five themes (lines 229-235). Please revise accordingly.

Response: Thank you very much for this observation. We have revised the sentence to portray 5 themes instead of four. (Page 11; Lines 247-249)

Review comment 5: Moreover, theme #3 (incoherence between description within instructions and actual contents of a test package) and theme #5 (diminished clarity in the wording of text instruction) conceptually overlaps so some redefining is needed. The choice to use the word “Contradictions” in the heading of section 3 of results (line 309) felt imprecise as the findings and quotes described in that section presented inconsistencies or lack of clarity in the instructions rather than contradictions. The term "incoherence" used previously is more precise.

Response: Thank you very much for your comment. We have used the word ‘inconsistencies’ instead of the word ‘contradictions’ in the theme #3. (Page 15; Line 327). In terms of overlaps between theme#3 and theme#5, we think that these two are different. Theme# is about what was written in word instruction being different from what people found in the packaged items. Theme#5 is about the word instruction being worded in a manner that users found it difficult to understand. We believe that these are two separate issues.

Reviewer #2 Comments

Review comment 1: Reviewer #2: Although Moses Kelly Kumwenda and coworkers' manuscript is in a field between scientific and social fields, it is of great interest, considering that rapid tests in low-literacy contexts could save lives.

Therefore, in the discussion section, the low number of participants is no problem. In my opinion, this is a driven study for future application on other rapid tests. In fact, my advice is to include the possibility of having an "optimizing instructional material" not only for COVID-19 rapid tests but for all commercially available self-sampling and testing tests.

Response: Thank you very much for the wonderful and though proving comments. Considering your comment on sample size, we have already acknowledged in the last sentence of the conclusion section that sample size differences did not have a bearing on the results (Page 26; Lines 582-584). We have also added a recommendation on having ‘optimisation materials for all commercially available self-sampling and testing products’ in the conclusion section. (Pages 26-27; Lines 595-597)

We will be looking forward to hearing from you soon.

Kind regards,

Moses Kumwenda (PhD)

On behalf of co-authors

---

## [Decision Letter · Decision Letter 1]

8 Nov 2024

TITLE: Optimising instructional materials for Covid-19 Rapid Tests for Self-Sampling and Testing: Mapping the optimisation process of manufacturers Instructions for Use for self-testing RDT's intended for low-literacy contexts

PONE-D-24-01686R1

Dear Dr. Kumwenda,

We’re pleased to inform you that your manuscript has been judged scientifically suitable for publication and will be formally accepted for publication once it meets all outstanding technical requirements.

Kind regards,

Bachti Alisjahbana, MD, PhD

Academic Editor

PLOS ONE

Additional Editor Comments (optional):

Dear Author

Your manuscript has importance in such that it would emphasis the need for better conduction of rapid test in primary care setting. Therefore I would suggest to accept this manuscript for further progress in Publication

Reviewers' comments:

Reviewer's Responses to Questions

**Comments to the Author**

1. If the authors have adequately addressed your comments raised in a previous round of review and you feel that this manuscript is now acceptable for publication, you may indicate that here to bypass the “Comments to the Author” section, enter your conflict of interest statement in the “Confidential to Editor” section, and submit your "Accept" recommendation.

Reviewer #1: All comments have been addressed

Reviewer #3: All comments have been addressed

2. Is the manuscript technically sound, and do the data support the conclusions?

Reviewer #1: (No Response)

Reviewer #3: Partly

3. Has the statistical analysis been performed appropriately and rigorously? 

Reviewer #1: (No Response)

Reviewer #3: N/A

4. Have the authors made all data underlying the findings in their manuscript fully available?

Reviewer #1: (No Response)

Reviewer #3: Yes

5. Is the manuscript presented in an intelligible fashion and written in standard English?

Reviewer #1: (No Response)

Reviewer #3: Yes

6. Review Comments to the Author

Reviewer #1: (No Response)

Reviewer #3: (No Response)

7. PLOS authors have the option to publish the peer review history of their article (what does this mean?). If published, this will include your full peer review and any attached files.

Reviewer #1: No

Reviewer #3: No

---

## [Editor Report · Acceptance letter]

14 Nov 2024

PONE-D-24-01686R1 

PLOS ONE

Dear Dr. Kumwenda, 

I'm pleased to inform you that your manuscript has been deemed suitable for publication in PLOS ONE. Congratulations! Your manuscript is now being handed over to our production team.

Kind regards, 

on behalf of

Dr. Bachti Alisjahbana 

Academic Editor

PLOS ONE